# Glutamate Transporters GltS, GltP and GltI Are Involved in *Escherichia coli* Tolerance In Vitro and Pathogenicity in Mouse Urinary Tract Infections

**DOI:** 10.3390/microorganisms11051173

**Published:** 2023-04-29

**Authors:** Hongxia Niu, Tuodi Li, Yunjie Du, Zhuoxuan Lv, Qianqian Cao, Ying Zhang

**Affiliations:** 1School of Basic Medical Sciences, Zhejiang Chinese Medical University, Hangzhou 310053, China; niuhongxia1985@163.com (H.N.);; 2School of Basic Medical Sciences, Lanzhou University, Lanzhou 730000, China; 3The First Affiliated Hospital, Zhejiang University School of Medicine, Hangzhou 310053, China

**Keywords:** uropathogenic *Escherichia coli*, tolerance, pathogenicity, glutamate transporter, glutamate

## Abstract

To verify the roles of GltS, GltP, and GltI in *E. coli* tolerance and pathogenicity, we quantified and compared the relative abundance of *gltS*, *gltP*, and *gltI* in log-phase and stationary-phase *E. coli* and constructed their knockout mutant strains in *E. coli* BW25113 and uropathogenic *E. coli* (UPEC) separately, followed by analysis of their abilities to tolerate antibiotics and stressors, their capacity for adhesion to and invasion of human bladder epithelial cells, and their survival ability in mouse urinary tracts. Our results showed that *gltS*, *gltP*, and *gltI* transcripts were higher in stationary phase *E. coli* than in log-phase incubation. Furthermore, deletion of *gltS*, *gltP*, and *gltI* genes in *E. coli* BW25113 results in decreased tolerance to antibiotics (levofloxacin and ofloxacin) and stressors (acid pH, hyperosmosis, and heat), and loss of *gltS*, *gltP*, and *gltI* in uropathogenic *E. coli* UTI89 caused attenuated adhesion and invasion in human bladder epithelial cells and markedly reduced survival in mice. The results showed the important roles of the glutamate transporter genes *gltI*, *gltP*, and *gltS* in *E. coli* tolerance to antibiotics (levofloxacin and ofloxacin) and stressors (acid pH, hyperosmosis, and heat) in vitro and in pathogenicity in mouse urinary tracts and human bladder epithelial cells, as shown by reduced survival and colonization, which improves our understanding of the molecular mechanisms of bacterial tolerance and pathogenicity.

## 1. Introduction

Bacterial infection has always been a threat to human survival, especially chronic and persistent relapse infections. Clinical examples include recurrent urinary tract infections with uropathogenic *Escherichia coli* [1], recalcitrant infections with *Mycobacterium tuberculosis* [2], and biofilm infections with *Pseudomonas aeruginosa* or *Staphylococcus aureus* [3]. In addition to antibiotic resistance, bacterial tolerance and persistence are recognized as other culprits for antibiotic treatment failure and the relapse of these prolonged and recurrent bacterial infections [4]. The ability of bacteria to survive exposure to high concentrations of antibiotics without acquiring resistance mutations is regarded as antibiotic tolerance. Unlike antibiotic resistance, which acquires genetic modifications, antibiotic tolerance results from non-genetic phenotypic changes in bacteria, usually associated with slow growth or reduced metabolism due to environmental stressors [5,6]. However, genetic mutations may also indirectly affect tolerance by altering metabolism, growth, or drug uptake [7]. Several pathways have been identified that are involved in antibiotic tolerance, including toxin-antitoxin (TA) modules [8] and stringent response via ppGpp [9], antioxidant defense [10], DNA repair [11], cell metabolism and energy production [12], trans-translation [13], and efflux [14]. In terms of metabolism, it has been found that genes that play roles in pathways of carbohydrate metabolism [15,16], phosphate metabolism [17], and amino acid and purine biosynthesis [18] are involved in antibiotic tolerance.

Previous studies have shown that glutamate metabolism plays a key role in bacterial survival under acid, hypoxia, and hyperosmotic stressors [19]. Specific examples include high expression of *gadC* (glutamate transporter-related gene) being important for survival of *E. coli* in acidic environments [20], *gltB* (glutamate synthase-related gene) and *gltC* (transcriptional regulator of *gltB*) in *Listeria monocytogenes* being involved in biofilm formation and antioxidant defense [21], and the glutamate-dependent acid-resistance system in *E. coli* being involved in protection against oxidative stress under extreme acid stress [22]. In addition, glutamate metabolism has been shown to be associated with *S. aureus* persistence [18,23]. These findings suggest that glutamate metabolism might be important in bacterial tolerance. However, it remains to be seen if glutamate transporter genes are involved in *E. coli* tolerance and whether the effects and mechanisms mediated via glutamate metabolism identified in vitro are related to *E. coli* pathogenicity in vivo. Among the L-glutamate transporters being identified in *E. coli*, the glutamate:γ-aminobutyrate exchanger GadC coupled with glutamate decarboxylase(s) GadA and GadB has been characterized as one of the most significant bacterial acid-resistant systems [24]. However, whether the other glutamate transporters GltS, GltP, and GltI are involved in *E. coli* tolerance and pathogenicity has yet to be determined.

Compared with other quinolones, tosufloxacin had higher activity against *E. coli* tolerant persister cells [25]. In our previous work to identify genes involved in *E. coli* persistence, we found that mutation in the *gltI* encoding glutamate/aspartate transport protein showed increased susceptibility to tosufloxacin [26], suggesting that glutamate transporters may be involved in bacterial tolerance. In this study, to evaluate the role of glutamate transporter GltS, as well as two other glutamate transporters, GltP and GltI, in *E. coli* tolerance and pathogenicity, we constructed their deletion mutants in both *E. coli* K12 strain BW25113 and uropathogenic *E. coli* UTI89, followed by analysis of their tolerance to various antibiotics and stressors in vitro and their pathogenicity in human bladder epithelial cells and mouse urinary tract infections.

## 2. Materials and Methods

### 2.1. Bacterial Strains and Cell Line

In this study, we used *E. coli* K12 strain BW25113 and uropathogenic *E. coli* strain UTI89, both of which were cultured in Luria-Bertani (LB) medium. Human bladder epithelial cells (ATCC HTB-9) were purchased from the Bio-feng company and cultured in RPMI 1640 with 10% newborn calf serum.

### 2.2. RNA Isolation, cDNA Construction, and Real-Time PCR Assay

*E. coli* BW25113 was cultured in LB broth to log phase (4 h) and stationary phase (12 h) separately for RNA isolation. RNA was extracted from the log phase and stationary phase bacteria using the GeneJET RNA Purification Kit (ThermoFisher Scientific, Waltham, MA, USA). Briefly, the cells were collected by centrifugation, disrupted with buffer supplemented with lysozyme (0.4 mg mL^−1^) and β-mercaptoethanol. Next, RNA in the lysate was purified using the GeneJET RNA Purification Column and eluted in nuclease-free water. The RNA preparations were quantified using Nanodrop (ThermoFisher Scientific) and treated with gDNA Eraser at 42 °C for 2 min prior to construct complementary DNA (cDNA).

Reverse transcription reactions were performed using the TaKaRa PrimeScript™ RT Kit (Takara Bio Inc., Kusatsu City, Japan). Real-time PCR was performed on the ABI StepOnePlus™ Real-Time PCR System using fluorescent reagent SYBR Green PCR Master Mix (TaKala, Dalian, China). The reaction system (20 μL) contained 2 × SYBR Green Premix (10 μL), primers F and R (2 μL each), cDNA template (2 μL), and sterile water (4 μL). After amplification, the temperature-dependent melting curves of the PCR products were examined by checking PCR specificity and product detection. The primers used were *gltS* A-F, *gltS* A-R, *gltP* A-F, *gltP* A-R, *gltI* A-F, *gltI* A-R, *rrsA* F, and *rrsA* R (Table 1). The cycle threshold (Ct) values of the reactions were obtained on Real-Time PCR Systems, and the relative mRNA expression level was determined by comparative 2^−ΔΔCt^ using *opgG* mRNA as the normalizer, where ΔΔCt = (Ct_glts/gltp/gltI_ − Ct_rrsA_)_staitionary phase_ − (Ct_glts/gltp/gltI_ − Ct_rrsA_)_log phase_. Triplicate samples of cells were collected at each time point and analyzed. 

### 2.3. Construction of GltS, GltP, and GltI Deletion Mutants in E. coli BW25113 and Uropathogenic E. coli UTI89

The deletion mutants Δ*gltS*, Δ*gltP*, and Δ*gltI* were constructed in *E. coli* BW25113 and uropathogenic *E. coli* UTI89 using the λ red homologous recombination system as described [27]. Briefly, *E. coli* BW25113 and uropathogenic *E. coli* UTI89 containing λ red recombinase helper plasmid pKD46 were generated by electroporation using a Bio-Rad Gene Pulser Xcell™ (Bio-Rad, Hercules, CA, USA). To construct linear FRTs (flippase recognition targets) on both sides of the chloramphenicol resistance gene for allelic exchange, PCR amplification was conducted using the pKD3 template and specific primers (Table 1), such as *gltS* B-F and *gltS* B-R for *gltS* deletion in *E. coli* BW25113, *gltS* C-F and *gltS* C-R for *gltS* deletion in *E. coli* UTI89; *gltP* B-F and *gltP* B-R for *gltP* deletion in *E. coli* BW25113, *gltP* C-F and *gltP* C-R for *gltP* deletion in *E. coli* UTI89, *gltI* B-F and *gltI* B-R for *gltI* deletion in *E. coli* BW25113, and *gltI* C-F and *gltI* C-R for *gltI* deletion in *E. coli* UTI89. Then, the purified amplified linear fragments were electroporated into L-arabinose-induced *E. coli* BW25113 and UTI89 cells harboring pKD46. Recombinant clones were selected on a medium containing chloramphenicol (25 μg/mL) and verified by PCR amplification. 

### 2.4. Complementation of E. coli Deletion Mutant Strains

To complement the *E. coli* BW25113 or UTI89 Δ*gltS,* Δ*gltP*, and Δ*gltI* mutant strains, recombinant plasmids pTrc99a-*gltS*, pTrc99a-*gltP*, and pTrc99a-*gltI* were constructed. Initially, the *gltS, gltP*, and *gltI* genes were cloned with primers *gltS* D-F and *gltS* D-R, *gltS P*-F and *gltP* D-R, and *gltI*-F and *gltI* D-R (Table 1). Then, the recombinant plasmids were transformed into their corresponding mutants to construct Δ*gltS-pTrc99a-gltS*, Δ*gltP-pTrc99a-gltP*, and Δ*gltI-pTrc99a-gltI* and control strains Δ*gltS-pTrc99a*, Δ*gltP-pTrc99a*, and Δ*gltI-pTrc99a*.

### 2.5. Evaluation of Tolerance of the Constructed Mutants to Antibiotics and Various Stressors

The abilities of tolerance to antibiotics and various stressors were evaluated by time-kill curve studies. Stationary phase *E. coli* BW25113, *E. coli* BW25113 Δ*gltS*, and its complemented strains were cultured overnight and treated with antibiotics in Eppendorf tubes, including levofloxacin (10 μM) and gentamicin (20 μM). Every day after treatment, the culture samples were taken out, washed, diluted, and plated for colony formation unit (CFU) count on LB agar plates. For acid, heat, and hypertonic stressors, stationary phase cultures were washed and exposed to corresponding conditions, i.e., LB medium with a pH 3.0 acidic condition for 5 days, LB medium supplemented with 3 M NaCl for 5 days, and a water bath in 52 °C heat condition for 6 h. Surviving bacteria were determined by CFU count daily or hourly.

### 2.6. Assays to Detect Bacterial Abilities to Adhere to and Invade Epithelial Cells

Human bladder epithelial cells (ATCC HTB-9) were used in these assays as described [28,29]. Briefly, cell suspension was prepared, and cells were seeded at 2 × 10^6^ cell/mL to a 24-well plate (1 mL per well) and infected with mid-log phase bacteria (MOI = 5~10). After co-incubation for 2 h at 37 °C, the epithelial cells were lysed with 0.1% Triton X-100, and the bacteria number in the lysates (represented as a) were counted by CFU counting. For adherence assay, the infected cells were washed and lysed, and the bacteria were counted (represented as b). Adherence frequency was the value of b/a. For invasion assay, the infected cells were washed and incubated with 100 μg/mL gentamicin for another 2 h. Then, the bacteria surviving incubation with gentamicin were counted (represented as c). Invasion frequency was the value of c/b.

### 2.7. Animals and Urinary Tract Infection Model in Mice

Female BALB/c mice aged 6–8 weeks were used in this study. The mice were purchased from Lanzhou Veterinary Research Institute and housed in a clean animal facility at Lanzhou University. Throughout the study, the mice received food and water freely, and the animal experiment protocols were approved by the Institutional Animal Protection and Use Committee of Lanzhou University (permit number: SYXK(Gan)2021–0305).

To analyze the pathogenicity of deletion mutants Δ*gltS*, Δ*gltP*, and Δ*gltI* in uropathogenic *E. coli* UTI89 in vivo, we used two UTI mouse urinary tract infection models in BALB/c mice as described [26,30]. One was using a stationary phase bacterial infection model [26,30], and the other was a biofilm bacterial infection model [26]. Briefly, mice were infected with 10^7^ CFU stationary phase bacteria or biofilm bacteria via the transurethral route. For stationary phase inoculum, *E. coli* UTI89 Δ*gltS*, Δ*gltP*, Δ*gltI*, and the parent strain were cultured for 24 h in LB broth without shaking. For biofilm inocula [31,32], *E. coli* UTI89 Δ*gltS*, Δ*gltP*, Δ*gltI*, and *E. coli* UTI89 were cultured for 24 h in LB broth in 96-well plates without shaking; the cultured planktonic bacteria were removed; and the biofilm bacteria in the bottom of the wells were scraped and resuspended in PBS before injection. After 1, 3, 5, or 6 days of infection, the bladders and kidneys of the infected mice were harvested, homogenized, serially diluted, and plated for CFU enumeration. 

### 2.8. Statistical Analysis

The results are expressed as means ± standard deviation (SD). Data were analyzed using one-way or two-way ANOVA and GraphPad Prism 9.0 software. Any *p*-value less than 0.05 is considered statistically significant.

## 3. Results

### 3.1. E. coli GltS, GltP, and GltI were Highly Expressed in the Stationary Phase

Previously, we found that the glutamate transporter gene *gltI* was involved in *E. coli* persister formation, where the *gltI* mutant showed decreased tolerance to tosufloxacin [26]. To assess the role of glutamate transporters in tolerance, which has not been reported before, we quantified the relative abundance of *gltI* and other glutamate transporter encoding genes *gltS* and *gltP* in stationary phase *E. coli*. Due to the depletion of nutrients in the medium and the accumulation of harmful metabolites during the stationary phase culture, the frequency of tolerant persisters is increased. Therefore, genes that are highly expressed in the stationary phase may be related to bacterial tolerance. In this study, the relative abundances of *gltS*, *gltP*, and *gltI* in stationary phase (12 h) *E. coli* were determined by real-time PCR assay, in which exponentially growing *E. coli* cells (4 h) were used as a control. The relative gene expression in stationary phase samples was determined by the 2^−ΔΔ*Ct*^ method using *rrsA* as the normalizer, where ΔΔ*Ct* = (Ct_targe gene_ − Ct_rrsA_)_staitionary phase_ − (Ct_targe gene_ − Ct_rrsA_)_log phase_. RT-PCR analysis revealed that the abundance of *gltS*, *gltP*, and *gltI* transcripts were increased two-fold from the log phase to the stationary phase transition (Figure 1).

### 3.2. Supplementation of Glutamate in Culture Medium Increased the Ability of E. coli Tolerance

To assess the role of glutamate transporters in bacterial tolerance, we first evaluated the effect of supplementation of glutamate in a culture medium on bacterial growth and tolerance. Compared with bacteria in a conventional culture medium, we found that supplementation of glutamate (≤1 mg mL^−1^) had no apparent effect on the growth of *E. coli* in both the log phase and the stationary phase (Figure 2a). After excluding the possibility that glutamate had an effect on bacterial growth, we cultured *E. coli* BW25113 in an LB medium with a relatively high concentration of glutamate (1 mg mL^−1^) and then evaluated its ability to alter tolerance to antibiotics (tosufloxacin, levofloxacin, and gentamicin) and stresses (acid pH, hyperosmosis, and heat conditions). The results showed that *E. coli* BW25113 cultured in LB medium supplemented with L-glutamate had higher tolerance to antibiotics and the environmental tested stressors than that in LB medium only (Figure 2b,c). These results indicated that glutamate was indeed involved in *E. coli* tolerance, further suggesting the role of glutamate transport proteins in this process.

### 3.3. Deletion of GltS, GltP, and GltI Genes in E. coli BW25113 Results in Decreased Tolerance to Antibiotics and Stress Conditions

To further verify the roles of *gltS*, *gltP*, and *gltI* in tolerance of *E. coli* to antibiotics and other stressors, we constructed *E. coli* BW25113 mutants Δ*gltS*, Δ*gltP*, and Δ*gltI* and their complemented strains. Prior to tolerance assay, we first performed growth curve studies with CFU counting to rule out the possibility of growth defects in *gltS*, *gltP*, and *gltI* mutants. We found that in both the log phase and the stationary phase, the growth abilities of *gltS*, *gltP*, and *gltI* deletion mutants under non-stress conditions were similar to those of the wild-type strain (Figure 3a and Figure 4a,b). We next performed an MIC experiment for levofloxacin and gentamicin with these three mutants. Compared with BW25113, Δ*gltS* had the same MIC for levofloxacin, and Δ*gltP* and Δ*gltI* showed a two-fold decrease in MICs for levofloxacin. All the deletion mutants showed the same gentamicin susceptibility as the parent strain (Table 2). The results showed that glutamate transporters GltS, GltP, and GltI had little or no effect on bacterial growth and susceptibility to antibiotics. 

We next performed the time-dependent tolerance assay, where stationary-phase bacteria of the BW25113 Δ*gltS*, Δ*gltP*, and Δ*gltI* mutants, their complemented strains, and the wild type were exposed to antibiotics including gentamicin (20 μM), levofloxacin (10 μM), and stress conditions, including acid pH (pH 3.0), hyperosmosis (3 M NaCl), and heat (52 °C) in vitro, and the surviving bacteria were determined. From the time-dependent killing curves, we found that the surviving bacteria of the deletion mutant Δ*gltS* decreased to the detection limit level 5 days after treatment with antibiotics (gentamicin and levofloxacin) and stressors (acid pH and hyperosmosis) or 6 h post treatment with heat condition, while the wild-type strain and its complemented strain Δ*gltS-pTrc99a-gltS* still had high levels of viable bacteria remaining, with more than 10^5^ CFU after gentamicin and levofloxacin treatment, more than 10^4^ CFU after acid pH or hyperosmosis treatment, and more than 10^3^ CFU after heat treatment (Figure 3b–f). The results showed that *gltS* is involved in the tolerance of *E. coli* to antibiotics (gentamicin and levofloxacin) and stress conditions (acid pH, hyperosmosis, and heat) in vitro.

Consistent with the *gltS* mutant, both *gltP and gltI* mutants also displayed significantly decreased tolerance to all the stressors in different assays. Due to limited space, we only show the surviving bacteria of Δ*gltP* and Δ*gltI*, their complemented strains, and the parent strain BW25113 at specific time points as treatment with levofloxacin and gentamicin for 3 days or exposure to hyperosmotic condition for 2 days, acidic condition for 2 days, or heat condition for 3 h (Figure 4). At these time points, the surviving bacteria of BW25113 and the complemented strains from treatments were maintained at 10^5^–10^6^ CFU, but the number of viable bacteria of the knockout mutant strains was 10–100 fold smaller (Figure 4). The results showed that deletion of *gltP* or *gltS* in *E. coli* resulted in significant (*p* < 0.01 or *p* < 0.001) decreased tolerance to antibiotics (gentamicin and levofloxacin) and stress conditions (acid pH, hyperosmosis, and heat), respectfully. 

### 3.4. Uropathogenic E. coli UTI89 Mutants ΔgltS, ΔgltP, and ΔgltI Exhibited Weakened Adhesion and Invasion to Human Bladder Epithelial Cells

In order to determine whether *gltS*, *gltP*, and *gltI* are involved in bacterial adhesion and invasion, we further analyzed the adhesion and invasion abilities of *E. coli* UTI89 Δ*gltS*, Δ*gltP*, and Δ*gltI* as well as the parental strain in human bladder epithelial cells. We found that both the adhesion and invasion rates of the Δ*gltS*, Δ*gltP*, and Δ*gltI* mutants were significantly lower than those of their parent strain *E. coli* UTI89 (*p* < 0.05) (Figure 5). The adhesion abilities of Δ*gltS*, Δ*gltP*, and Δ*gltI* mutants were at least 100 times lower than those of *E. coli* UTI89 (Figure 5a), and the invasion rates of *E. coli* UTI89 Δ*gltS*, Δ*gltP*, and Δ*gltI* were decreased 2~100 times compared with *E. coli* UTI89 (Figure 5b). The results suggest that *gltS*, *gltP*, and *gltI* are important for the adhesion and invasion abilities of uropathogenic *E. coli.*

### 3.5. Loss of GltS, GltP, and GltI in Uropathogenic E. coli UTI89 Caused Markedly Reduced Survival in Mice

To further assess the pathogenicity of the *gltS*, *gltP*, and *gltI* mutants in vivo, we constructed Δ*gltS*, Δ*gltP*, and Δ*gltI* mutants in uropathogenic *E. coli* UTI89 and evaluated their pathogenicity in BALB/c mice. The mice were infected with stationary-phase or biofilm bacteria of *E. coli* UTI89 Δ*gltS*, Δ*gltP*, and Δ*gltI* and the wild-type strain, then the bacterial loads in the bladders and kidneys of infected mice were measured on days 1, 3, 5, and 6 after infection, respectively. For different strains, the bacterial loads in the bladders and kidneys of the mice receiving stationary phase bacteria decreased gradually over time in general. But the UTI89 deletion mutants Δ*gltS*, Δ*gltP*, and Δ*gltI* had significantly lower bacterial loads than those of the parent strain UTI89 at all the time points post infection (*p* < 0.01) (Figure 6a,b). In the bladder (Figure 6a), on the fifth day post infection, mice infected with the *gltP* and *gltI* mutants had no bacteria left, whereas mice infected with the UTI89 strain had 10^4^ CFU remaining. In the kidneys (Figure 6b), no bacteria were detected in the *gltS* mutant-infected mice on the first day post infection, whereas mice infected with the UTI89 strain still had 10^5^ CFU bacteria; on the third day post infection, the bacterial load in mice infected with the *gltP* and *gltI* mutants decreased to a level below the detection limit, whereas there were 10^3^ CFU bacteria in mice infected with the parent strain UTI89. Three days after infection, in both the bladder and kidneys, all the complemented strains restored the pathogenicity of the deletion mutants (Figure 6c,d).

Biofilm bacteria are more tolerant than stationary phase bacteria. In the biofilm infection mouse model, at 6 days post infection, mice infected with the mutant strains Δ*gltS*, Δ*gltP*, and Δ*gltI* contained 10^3^~10^6^ CFU in the bladder, kidneys, or urine, whereas mice infected with the parent strain UTI89 had ~10^8^ CFU bacteria (*p* < 0.01) (Figure 6e). These results indicate that the *gltS*, *gltP*, and *gltI* mutants survive less well in vivo than the parental strain UTI 89.

## 4. Discussion

Glutamate plays an important role in various metabolic processes in bacterial cells. Among the L-glutamate transport systems in *E. coli*, GadC has been identified as a glutamate antiporter in the cytoplasmic membrane and plays a role in regulating acid resistance [33]. In this research, our results showed that the deletion mutants of glutamate transporters Δ*gltS*, Δ*gltP*, and Δ*gltI* had decreased tolerance to several stressors including antibiotics such as levofloxacin and gentamicin and other conditions such as acid pH, hyperosmosis, and heat. In addition, the survival and colonization abilities of the glutamate transporter mutants in mice and adhesion to and invasion of human bladder epithelial cells were also reduced (Figure 3, Figure 4 and Figure 5). These results suggest that the glutamate metabolism and transporter genes *gltS*, *gltP*, and *gltI* are involved in *E. coli* tolerance to antibiotics (levofloxacin and ofloxacin) and stressors (acid pH, hyperosmosis, and heat) and pathogenicity. These findings indicate that glutamate metabolism and transport could serve as targets for developing new drugs against persister bacteria.

It has been recently reported that antibiotic-tolerant *E. coli* cells have a lower intracellular pH than susceptible cells, and bacteria upregulates glutamate decarboxylases (GadA) to counteract this intracellular acidification [34], which also indicates the important role of glutamate for bacteria in surviving antibiotic exposure and stressors. There are several possible mechanisms by which glutamate may mediate tolerance to antibiotics (levofloxacin and ofloxacin) and stressors (acid pH, hyperosmosis, and heat). The first is acid tolerance. Bacteria have developed a wide range of strategies to overcome acid stress, where the glutamate-dependent acid-resistance system is the most effective defense mechanism in protecting cells from exposure to low pH environments [24,35]. For instance, the decarboxylation of glutamate by two glutamic acid decarboxylases (encoded by *gadA* and *gadB*) consumes protons and therefore removes intracellular protons from the acidic conditions [24,35]; the byproduct of the glutamate decarboxylation γ-aminobutyrate (GABA) is exported by the antiporter *gadC*, which performs the glutamate_in_ and GABA_out_ [36] and also helps overcome acid stress by increasing the pH. Second, glutamate and its metabolite GABA are prominent compatible solutes in bacteria that protect enzymes and enable them to function efficiently under multiple stressors, including high temperature, freeze-thaw treatment, or drying [37]. Third, glutamate may help bacteria to tolerate antibiotics or stressors through affecting energy metabolism. L-glutamate is the only amino acid that performs oxidative decarboxylation at a fairly high rate. Under nutrition limitation, glutamate dehydrogenase activated by GDP and ADP converts glutamate to α-ketoglutaric acid (α-KG) through oxidation and deaminization, which brings glutamate into the energy metabolism via the TCA cycle, which then facilitates bacterial tolerance [38]. 

Given the important role of glutamate in *E. coli* tolerance to antibiotics (levofloxacin and ofloxacin) and stressors (acid pH, hyperosmosis, and heat), we suppose that pathogenic *E. coli* glutamate transporters might hijack glutamate to enhance its survival ability and pathogenicity in vivo. To test this, we constructed deletion mutants of Δ*gltS*, Δ*gltP* and Δ*gltI* in uropathogenic *E. coli* UTI89 and evaluated their survival ability in a mouse model of a urinary tract infection. From day 1 to day 6 after the infection, we found that the Δ*gltS*, Δ*gltP*, and Δ*gltI* mutants had reduced survival and colonization abilities compared with the wild-type strain (Figure 6). Based on the results of in vitro tolerance assays, it is conceivable that uropathogenic *E. coli* could take in glutamate using these transporters like GltS to overcome stressors in urine, including low pH, poor medium, and frequently changing osmolarity, hence increasing its ability for survival and colonization in the urinary tract. One reason may be that uropathogenic *E. coli* uses glutamate to overcome stressors in urine including low pH, poor medium, and frequently changing osmolarity, as described above. In addition, uropathogenic *E. coli* might also evolve the glutamate metabolic pathway to interfere with host cell metabolism and hence lead to a suboptimal innate immune response to enhance its persistence ability in the host [39]. Further studies are needed to verify this possibility. Furthermore, we found that the Δ*gltS* mutant loses the flagellar rotor protein FliG in our proteomic analysis (unpublished observation), which may lead to decreased capabilities of adhesion to and invasion of human bladder epithelial cells and hence reduce colonization.

Both levofloxacin and gentamicin are clinical drugs that treat pathogenic *E. coli*. This research was started based on our previous finding that *E. coli* mutation in the *gltI* encoding glutamate/aspartate transport protein showed increased susceptibility to tosufloxacin. So, we chose two quinolones (levofloxacin and tosufloxacin) to evaluate whether the decreased tolerance of the deletion mutants was specific to tosufloxacin, and we chose one aminoglycoside (gentamicin) to evaluate whether it is specific to quinolones. From the results, we found that *E. coli* genes encoding glutamate transporters are not specific to tosufloxacin but are also involved in its tolerance to other categories of antibiotics and stressors. These findings are consistent with our previous findings that *S. aureus* mutations in genes associated with glutamate metabolism are defective in persistence or tolerance to rifampicin [18].

## 5. Conclusions

In summary, we identified the important roles of glutamate transporter genes *gltI*, *gltP*, and *gltS* in *E. coli* tolerance to several antibiotics and stressors in vitro as well as their effects in UPEC pathogenicity in mouse urinary tracts, including their ability to survive and colonize in the bladder and kidneys and suppress pro-inflammatory cytokines. Our findings improve our understanding of the molecular mechanism of bacterial tolerance and pathogenicity and suggest that novel drugs that target or inhibit glutamate transport may be helpful for treating persistent UPEC infections.

## Figures and Tables

**Figure 1 microorganisms-11-01173-f001:**
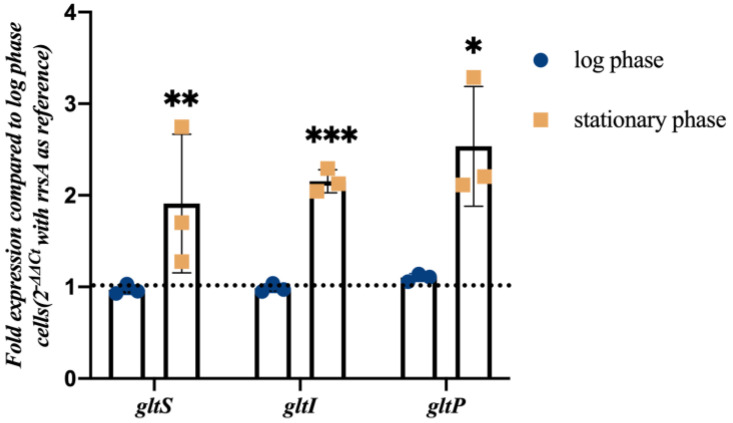
The expression of *gltS*, *gltp*, and *gltI* genes in stationary phase *E. coli* compared to that in log phase cells. *E. coli* BW25113 was cultured to log phase (4 h) and stationary phase (12 h) separately for RNA isolation. The relative gene expression was determined using the comparative 2^−ΔΔ*Ct*^ method using *rrsA* as the normalizer, where ΔΔ*Ct* = (Ct_target gene_ − C_rrsA_) _stationary phase_ − (Ct_target gene_ − Ct_rrsA_) *log phase*, target gene indicates *gltS*, *gltP* or *gltI*. Triplicate samples of cells were collected at each time point. * *p* < 0.05, ** *p* < 0.01, *** *p* < 0.001, relative to log phase.

**Figure 2 microorganisms-11-01173-f002:**
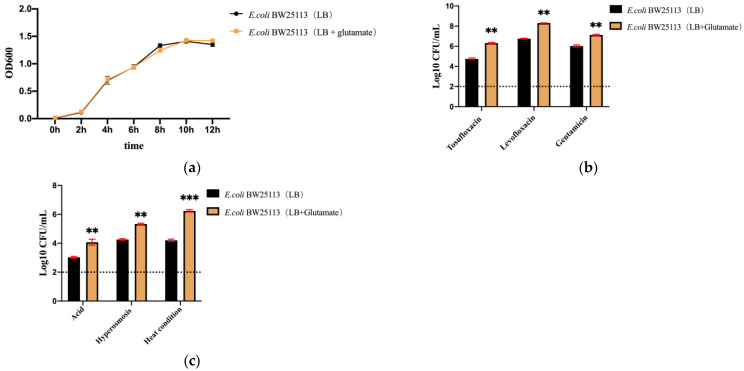
Growth and tolerance ability of *E. coli* cultured in medium supplemented with glutamate. (**a**) Growth curve of *E. coli* BW25113 in LB or LB + glutamate (1 mg/mL). (**b**) Tolerance of *E. coli* cultured in LB medium supplemented with glutamate to antibiotics. *E. coli* BW25113 was cultured overnight and then exposed to different antibiotics tosufloxacin (10 μM), levofloxacin (10 μM), and gentamicin (20 μM). Surviving bacteria were counted on the third day post treatment by CFU count. (**c**) Tolerance of *E. coli* cultured in LB medium supplemented with glutamate to stress conditions. The overnight cultures were exposed to a pH 3.0 condition for 2 days, a 3 M NaCl condition for 2 days, or a 52 °C water bath for 3 h. At different time points, surviving bacteria were counted by CFU count. The results are expressed as means ± SD. ** *p*-value < 0.01, *** *p*-value < 0.001, relative to bacteria cultured in conventional LB only.

**Figure 3 microorganisms-11-01173-f003:**
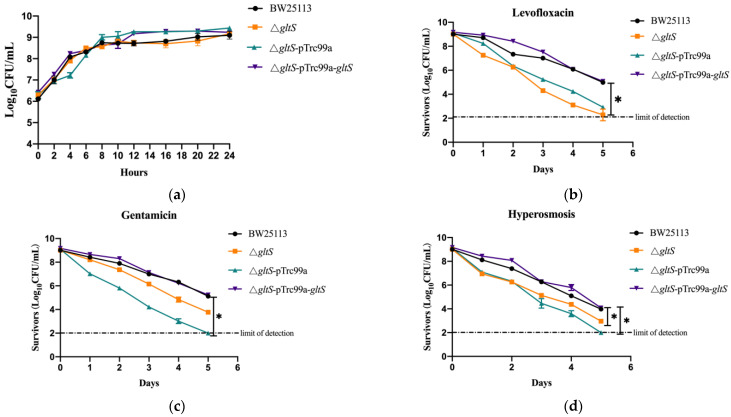
The tolerance ability of *E. coli* BW25113 Δ*gltS* mutant and its complemented strain to antibiotics and stressors. (**a**) Growth curves of *E. coli* BW25113, *E. coli* BW25113 Δ*gltS*, and its complement strain. (**b**,**c**) Antibiotic killing curves. Bacteria cultured overnight were exposed to antibiotics levofloxacin (10 μM) and gentamicin (20 μM) separately. Surviving bacteria post treatment were counted daily by CFU counting. (**d**–**f**) Survival of the stationary phase bacteria to stress conditions. The overnight cultures were exposed to acidic pH (pH 3.0), hyperosmosis (3 M NaCl), and heat (52 °C) conditions separately. Surviving bacteria post exposure at different times were determined by CFU count. * *p*-value < 0.05.

**Figure 4 microorganisms-11-01173-f004:**
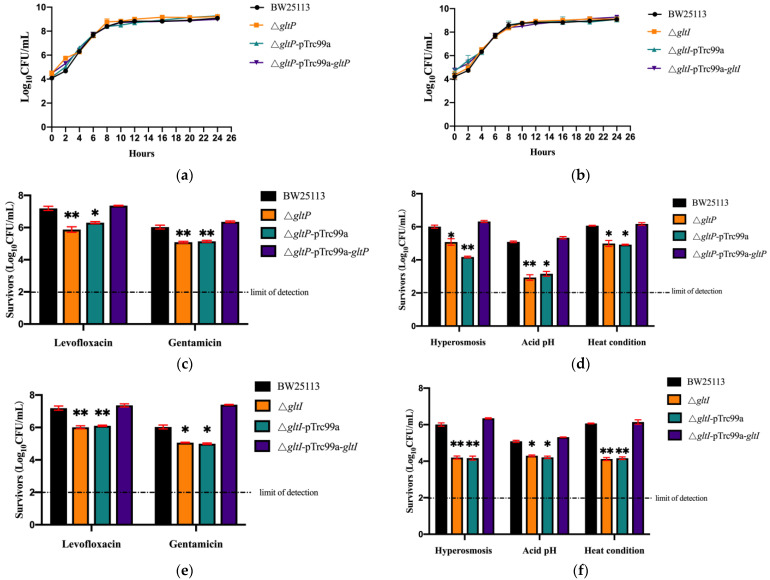
Survival of *E. coli* BW25113 Δ*gltP* and Δ*gltI* mutants and their complemented strains under antibiotics and stressors. (**a**,**b**) The growth curves of *E. coli* BW25113, *E. coli* BW25113 Δ*gltP*, and its complemented strain. (**c**,**e**) The ability of bacteria to tolerate antibiotic killing. The bacteria strains were cultured overnight and then exposed to antibiotics levofloxacin (10 μM) and gentamicin (10 μg/mL). Surviving bacteria were counted at the third day post treatment by CFU count. (**d**,**f**) The ability of bacteria to tolerate stress damage. The overnight cultures were exposed to acidic condition of pH 3.0 for 2 days, heat condition of 52 °C for 3 h, or hyperosmotic condition of 3 M NaCl for 2 days. At different time points, surviving bacteria were counted by CFU count. * *p* < 0.05, relative to *E. coli* BW25113 group. ** *p* < 0.01, relative to *E. coli* BW25113 group.

**Figure 5 microorganisms-11-01173-f005:**
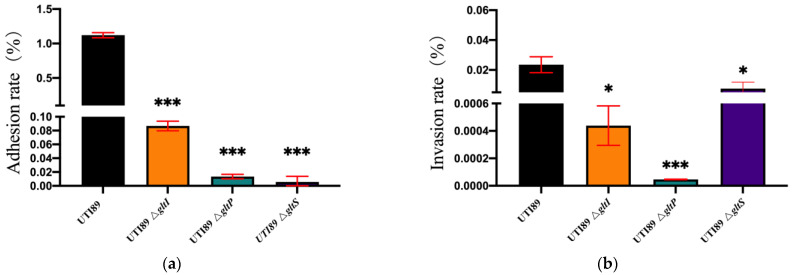
The adhesion and invasion abilities of uropathogenic *E. coli* UTI89 mutants Δ*gltS*, Δ*gltI*, and Δ*gltP* to bladder epithelial cells. The human epithelial cells (2 × 10^6^ cells/well) were infected with *E. coli* UTI89 and its mutants Δ*gltS*, Δ*gltP*, and Δ*gltI* at a multiplicity of infection (MOI) of 5~10 for 2 h in 24-well plates. Then, the frequencies of bacteria that adhere to and invade the cells were analyzed. The results are expressed as means ± SD. * *p* < 0.05, *** *p* < 0.001, relative to *E. coli* UTI89 group. (**a**) The adhesion rate; (**b**) The invasion rate.

**Figure 6 microorganisms-11-01173-f006:**
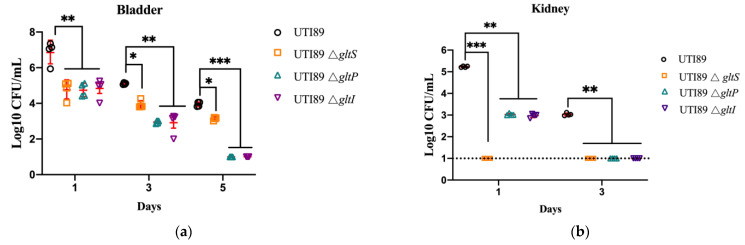
The colonization of uropathogenic *E. coli* UTI89 Δ*gltS*, Δ*gltP*, and Δ*gltI,* their complemented strains, and the wild-type strain in urinary tracts of Balb/c mice. (**a**) The CFU of colonized bacteria in the mouse bladder infected with stationary-phase UTI89 and its mutants Δ*gltS*, Δ*gltP*, and Δ*gltI*. (**b**) The CFU of colonized bacteria in the kidney of the mice infected with stationary-phase UTI89 and its mutants Δ*gltS*, Δ*gltP*, and Δ*gltI* for different days. (**c**) The CFU of colonized bacteria in the mouse bladder infected with stationary-phase UTI89, its mutants Δ*gltS*, Δ*gltP*, and Δ*gltI* and their complementary strains for 3 days. (**d**) The CFU of colonized bacteria in the kidney of the mice infected with stationary-phase UTI89, its mutants Δ*gltS*, Δ*gltP*, and Δ*gltI* and their complementary strains for 3 days. (**e**) The CFU of colonized bacteria in the bladders, kidneys, and urine of the mice infected with biofilm UTI89 for 6 days. The mice were infected with *E. coli* UTI89, Δ*gltS*, Δ*gltP*, and Δ*gltI* mutants (10^7^ CFU per mice) via the transurethral route, respectively. On days 1, 3, and 5 post infection, mice were sacrificed and the bacterial loads in the bladder and kidneys were counted. * *p* < 0.05, *** p* < 0.01, *** *p* < 0.001, relative to *E. coli* UTI89 group.

**Table 1 microorganisms-11-01173-t001:** Primer sequences for amplifying the *gltS*, *gltI*, and *gltP* genes. The italicized sequences are restriction sites. The underlined sequences are flippase recognition targets (FRTs) flanked by chloramphenicol resistance genes.

Primer Name		Sequence (5′-3′)
For RT-PCR	** *gltS* ** ** A-F**	GTTCATTGAACGTTATGGCTTC
	** *gltS* ** ** A-R**	CCGCCAATCAAGCCGCCCAG
	** *gltP* ** ** A-F**	GCAGTTCCCACGGCATTATG
	** *gltP* ** ** A-R**	CAGAGATGGAGCGGAACACG
	** *gltI* ** ** A-F**	CGATTTTGAATGTGGTTCTAC
	** *gltI* ** ** A-R**	GACTACGGCTTTGTCTTTCAG
	** *rrsA* ** **-F**	GAAAGGGGAGTGGGGTAAAGG
	** *rrsA* ** **-R**	CGGCTGAAGGTGATGGTGT
For deletion mutant construction in *coli* BW25113	** *gltS* ** ** B-F**	GATGAAGCGGCGGTAGAAGTGCCGCCGCAACAAAGACAAATGCCTGATGTGTAGGCTGGAGCTGCTTC
	** *gltS* ** ** B-R**	ATCAGGCATTTGTCTTTGTTGCGGCGGCACTTCTACCGCCGCTTCATCGGTATGGGAATTAGCCATGGTCC
	** *gltP* ** ** B-F**	TTCTCGCGTTTCTGAACGGGGAACGGCGCTCCATTGAGGAAGTTATTCTGGTGTAGGCTGGAGCTGCTTC
	** *gltP* ** ** B-R**	AGTCAGGCATCCACACATTGCCGGGTGGATATCCCCCGGCAATCTTCAAATGGGAATTAGCCATGGTCC
	** *gltI* ** ** B-F**	TCACAACGGGTATCCATGCGTTCTTAACGCAGAAGATAAAGGAGTTGGATGTGTAGGCTGGAGCTGCTTC
	** *gltI* ** ** B-R**	TGCTACGTAACAATCGAGAGGGCTGGAATTTCCGCCCCTGGTTCTTGTAAATGGGAATTAGCCATGGTCC
For deletion mutant construction in *E. coli* UTI89	** *gltS* ** ** C-F**	GTTACTCGAATGCGTAAAAAGCGGCGGTGAGAAGACCGCCGCTTCATCGGGTGTAGGCTGGAGCTGCTTC
	** *gltS* ** ** C-R**	GATGAAGTATGACGAGTATGAAAGAGTGATGCGGACACAAAGGAGTAACTATGGGAATTAGCCATGGTCC
	** *gltP* ** ** C-F**	TTCTCGCGTTTCTGAACGGGGAACGGCGCTCCATTGAGGAAGTTATTCTGGTGTAGGCTGGAGCTGCTTC
	** *gltP* ** ** C-R**	AGTCAGGCATCCACACATTGCCGGGTGGATATCCCCCGGCAATCTTCAAATGGGAATTAGCCATGGTCC
	** *gltI* ** ** C-F**	TGCTACGTAACAATCGAGAGGGCTGGAATTTCCGCCCCTGGTTCTTGTAAGTGTAGGCTGGAGCTGCTTC
	** *gltI* ** ** C-R**	TCACAACGGGTATCCATGCGTTTTTTAACGCAGAAGATAAAGGAGTTGGATATGGGAATTAGCCATGGTCC
For complementation of deletion mutants	** *gltS* ** ** D-F**	CG*GAATTC*ATGTCGATACTTTAGCAACGCTTGT (*Ecor* I)
	** *gltS* ** ** D-R**	TCG*AAGCTT*TTACCAGCGCATTGACGATA (*BamH* I)
	** *gltP* ** ** D-F**	CCG*GAATTC*ATGCGTAACGAACTGAACGG (*Eco*R I)
	** *gltP* ** ** D-R**	CGC*GGATCC*TTATTTTCAATCAACTGGATCAGG (*Bam*H I)
	** *gltI* ** ** D-F**	GG*AATTC*ATGCAATTACGTAAACCTGC (*Eco*R I)
	** *gltI* ** ** D-R**	CGG*GATCC*TTAGTTCAGTGCCTTGTC (*Bam*H I)

**Table 2 microorganisms-11-01173-t002:** The MICs of levofloxacin, gentamicin, and tosufloxacin against *E. coli* BW25113 Δ*gltS*, Δ*gltP*, and Δ*gltI* mutants.

Strains	Levofloxacin (μM)	Gentamicin (μM)	Tosufloxacin (μM)
BW25113	0.03	2.75	0.03
Δ*gltS*	0.03	2.75	0.015
Δ*gltP*	0.015	2.75	0.0075
Δ*gltI*	0.015	2.75	0.03

## Data Availability

All datasets generated for this study are included in the article.

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
