# Peer review of "Glutamate Transporters GltS, GltP and GltI Are Involved in Escherichia coli Tolerance In Vitro and Pathogenicity in Mouse Urinary Tract Infections"

_microorganisms, 2023, doi:10.3390/microorganisms11051173_

Round 1

Reviewer 1 Report

In this paper the authors correlate the role of Glutamate Transporter GltS, GltP and GltI with  Escherichia coli tolerance in vitro, in E. coli  BW25113 and with pathogenicity in mouse  in E. coli (UPEC) UTI89, studyng the transcripts of relatives genes (gltS, gltP and gltI).  Authors demonstrate that gene trascripts were higher in stationary phase than in log phase and that the deletion of these genes in E. coli BW25113 and in uropathogenic E coli result, respectivey in decreased tolerance to stress condition and antibiotic (BW25113) and in attenuated  pathogenicity in mouse (UTI89).

Although previous studies have shown that glutamate metabolism plays a key role in bacterial survival under acid, hypoxia and hyperosmotic stresses and have been associated with S. aureus persistence , Hongxia et al. investigate if glutamate transporter genes are involved in E. coli tolerance and whether the effects and mechanisms mediated via glutamate metabolism, identified in vitro are related to E. coli pathogenicity in vivo. The results obtained by authors   are very interesting and surely give a good contribute to shed further light about the correlation of glutamate transport and virulence in vitro. Methods are also consistent and appropriate.

 Comment.

I do not understand why the authors which masterfully demonstrate the decresed tolerance in deleted E.coli BW25113 versus wild type and complemented strains, do not show any complementation in deleted uropathogenic E. coli UTI89 strains to demonstrate just as masterfully the decrease virulence of the deleted strains in vivo. Moreover, in methods the same authors declare the construction of complemented UTI89 strains. In my opinion, showing a complementation by complemented UTI89 strains, exactly as the authors did for BW25113, would give a stronger evidence that the genes gltS, gltP and gltI, involved in glutamate transport are also really involved in the virulence. 

Author Response

Comments: I do not understand why the authors which masterfully demonstrate the decresed tolerance in deleted E. coli BW25113 versus wild type and complemented strains, do not show any complementation in deleted uropathogenic E. coli UTI89 strains to demonstrate just as masterfully the decrease virulence of the deleted strains in vivo. Moreover, in methods the same authors declare the construction of complemented UTI89 strains. In my opinion, showing a complementation by complemented UTI89 strains, exactly as the authors did for BW25113, would give a stronger evidence that the genes gltS, gltP and gltI, involved in glutamate transport are also really involved in the virulence. 

Response: Thanks for your helpful and constructive comments. We agree with your opinion that showing complementation of the UTI89 mutant strains would give a stronger evidence that the genes gltS, gltP and gltI involved in glutamate transport are involved in virulence. Although we constructed complemented strains of the UTI89 ΔgltS, ΔgltP and ΔgltI mutants, due to the limited number of animals available, we did not include them in the experiment for virulence evaluation. We could evaluate their virulence in future experiments.

Reviewer 2 Report

Introduction: highlight the importance of tosufloxacin especially compared with other fluoroquinolones, for example as persister treatment, to justify the rationale of the experimental design.

Two quinolones and 1 aminoglycoside were used to test the strain, why is the reason for those? Please explain.

Section 2.3 Table 1: information from table will be easier to addresses if the primers were categorized based on the experiment for which they were designed, i.e., adding an extra merged column explaining if qPCR, mutant construction...

Subsection 2.6. All the conditions should be better explained (line 139, rephrase "i.e., [...]").

Have any of the 3 proposed genes expression tested on the experimental conditions?

Section 3.3. Authors conclude that there is no difference in MICs while that is not what is shown in Table 2.

In general, authors mention that these genes have a role in "antibiotic resistance or tolerance". I suggest to specify to the "tested antibiotics", as well as the environmental stresses that were tested.

Figure 3: significance is missing, please run statistics on the growth curves.

Figure 3: Why tosufloxacin is not included? Please specify or perform the corresponding assays.

Lines 260-261: which burden corresponde to each conditions? Please rephrase.

Why the difference of the graphs and assays between Figure 3 and 44, for the different mutants?

Figure 4. Please add proper label for significance. For example, the used bars in in c or d are showing that the significance difference is between the green and orange. Add individual labels and bars comparing each column with the control. It was also once-way anova performed in here? Again, tosufloxacin is missing, why?

In all the figures as well as in method sections authors says that data is expressed as mean +-SD, while in most of the bar graphs only above bars are shown, please fix.

Figure 6c, at which point after infection were those samples taken?

Discussion section also should be improved highlighting the importance and relevance of the obtained results.

Minor

Subsection 2.1 and 2.8 can be merged

Line 144: cell suspension was prepared and cells seeded at [...].

Line 146: for 2 hours

Line 160: inocula

Line 164: remove one "after"

Line 165: "homogenized, serially diluted, and plated for CFU enumeration".

Line 168: is without bold.

Line 202: "and the environmental tested stresses"

Figures are cite in the text with capital later but in the pictures with lower case, please standardize.

"BALB/c mice" along the text

Round 2

Reviewer 1 Report

Major Comment

Sorry but I have to insist about introducing complemented strains of the UTI81 in vivo experiments. At least to demonstrate   survival in mice. The lack of  little number of mice available is not a good or convincent reason to exclude such important strains from in vivo experiment.

Author Response

Response: Thanks for your constructive comments. We have introduced the complementation strains of the UTI89 deletion mutants in the urinary tract infection experiment in Balb/c mice, and found that three days after infection, in both bladder and kidney, all the complemented strains restored the survival of the deletion mutants. The results have been added to the revised manuscript as Fig. 6c and d as shown in the attachment. We hope this revision will meet your request. 

Reviewer 2 Report

Authors answered all my previous comments and inquiries.

Author Response

Thank you again for you helpful review.

Round 3

Reviewer 1 Report

Ok. Thank you for answered to my suggestion